# Multi-packet transmission aero-engine DCS neural network sliding mode control based on multi-kernel LS-SVM packet dropout online compensation

Li Guangfu[1,2]*, Wang Xu[1,3], Ren Jia[1,4]

**1** Data Science Institute of City University of Macau, Macau, China, **2** Department of Network Information of Shandong University of Art & Design, Jinan, Shandong, China, **3** College of Business of Lingnan Normal University, Zhanjiang, Guangdong, China, **4** College of Information and Technology of Hainan University, Haikou, Hainan, China

\* ligf@sdada.edu.cn

**Data Availability Statement:** All relevant data is within the paper and its Supporting Information files.

## Abstract

In view of the strong nonlinear characteristics of the multi-packet transmission Aero-engine DCS with induced delay and random packet dropout, a neural network PID approach law sliding-mode controller using sliding window strategy and multi-kernel LS-SVM packet drop-out online compensation is proposed. Firstly, the time-delay term in the system model is transformed equivalently, to establish the discrete system model of multi-packet transmission without time-delay; furthermore, the construction of multi-kernel function is transformed into kernel function coefficient optimization, and the optimization problem can be solved by the chaos adaptive artificial fish swarm algorithm, then the online predictive compensation will be made for data packet dropout of multi-packet transmission through the sliding window multi-kernel LS-SVM. After that, a sliding-mode controller design method of proportional integral differential approach law based on neural network is proposed. And online adjustment of PID approach law parameters can be achieved by nonlinear mapping of neural network. Finally, Truetime is used to simulate the method. The results shows that when the packet dropout rate is 30% and 60%, the average error of packet dropout prediction of multi-kernel LS-SVM reduces 29.21% and 44.66% compared with that of combined kernel LS-SVM, and the chattering amplitude of the proposed neural network PID approach law sliding-mode controller is decreased compared with other five approach law methods respectively. This controller can ensure a fast response speed, which shows that this method can achieve a better tracking control of the aeroengine network control system.

## Introduction

Distributed control system stands out for its unique advantages in structure, controllability and reliability, which represents the developing orientation of the aero-engine control system in 21st century [1].By using distributed sensor, actuator and bus network, the weight of

**Funding:** International Science and Technology Cooperation Projects of China (2015DFR10510), National Natural Science Foundation of China and Macau Science and Technology Development Joint Fund (NSFC-FDCT)(6191101101), National Natural Science Foundation of China (61562018), Key Science and Technology Projects of Haikou, Hainan Province (2017041),Hainan Provincial Natural Science Foundation of China (519QN180), Hainan Provincial Key R & D Plan (ZDYF2019014), National Natural Science Foundation of China and Macau Science and Technology Development Joint Fund (0066/2019/AFJ), the program of and the Scientific Research Foundation of Hainan University (KYQD(ZR)1859) are gratefully acknowledged.

**Competing interests:** The authors have declared that no competing interests exist.

aeroengine control system is greatly reduced, the control performance and reliability are greatly improved, the development cost is significantly reduced, and the development cycle is greatly shortened. Distributed control system is essentially a kind of networked control system [2]. However, there are some inevitable problems in NCS, among which the most influential ones are node-driven mode, network-induced delay, packet dropout, communication restriction, multi packet transmission, and so on [3].

When there is time-delay in NCS, the commonly used method is to treat network delay as model uncertainty or use model equivalent transformation to eliminate the effect of delay [4,5]. Due to the advantages of sliding-mode control, there has been some research results achieved about its application in time-delay system and time-delay network control system [6–8]. In view of the time-delay in the system, the conventional design idea of the existing research results is to make full use of the invariance of the sliding-mode control, to design a sliding surface satisfying the progressive stability condition by LMI [9], free weight matrix [10] and other methods, and to improve both the approach law and the controller design [11,12] so that ensuring the stability and robustness of the system.

There are two main methods for solving data packet dropout, one is to convert packet dropout into uncertain long time-delay [13], the other is to use intelligent algorithms such as neural network and support vector machine as compensators to predict and compensate the packet dropout [14]. In reference [15], the combined kernel function SVM is used to predict and compensate the packet dropout, and good results are achieved. For the packet dropout in single packet transmission system, a common design idea for sliding-mode controller is to define a robust sliding surface based on a certain compensation strategy, and then further design a sliding-mode controller meeting the reachability condition [16,17]. Another mature idea is to use the multi-step prediction method to deal with the packet dropout in the network. The prediction results are used in the design process of the sliding-mode controller to ensure the system stability [18,19].

However, in the actual NCS, because of the limitation of the maximum allowable data frame capacity, and the wide distribution of sensors or controller nodes, combining and packaging the data of multiple nodes will undoubtedly increase the design cost of the system, so the data would be transmitted between different nodes in the way of multiple data packets [20,21]. Multi-packet transmission brings new problems to the design of network control system [22,23]. Due to the bandwidth limitation of communication network, the state quantity or control quantity that is divided into multiple data packets for transmission cannot reach the controller or actuator node at the same time, resulting in only a part of the controller or actuator variables can be updated in time [24]. However, the current packet dropout researches in the references are all carried out under the condition of single packet transmission, without considering multi-packet transmission, and the construction of kernel function will have a great impact on the final prediction accuracy of the support vector machine [25]. Now, empirical method or trial-and-error method is usually adopted when constructing kernel function, all of these methods mentioned above have certain performance conservatism [26].

Sliding-mode variable structure control has been widely used in the field of nonlinear control because of its excellent robustness [27]. In reference [28], an integral sliding-mode variable structure controller is proposed, which uses the load torque observer to suppress the effect of load disturbance. In reference [29], a control method of complementary sliding-mode variable structure is proposed. By combining the complementary sliding surface with the generalized sliding surface, a better control effect is obtained. In reference [30], the variable index approach law is applied to sliding-mode variable structure control, which can effectively reduce the chattering of the system at one time. However, these papers mentioned above only consider how to reduce chattering, without taking the overall performance optimization of

sliding-mode control into consideration, and there is no influence of packet dropout and time-delay on the research objects [31].

Considering the strong nonlinear characteristics between different actual systems and the adverse effects of network transmission on the control system, this paper proposes a neural network PID approach law sliding-mode control method of packet dropout online compensation multi-packet transmission network control system, which is based on the combination of sliding time window optimized by chaos artificial fish swarm algorithm and multi-kernel LS-SVM.

The main contributions of this paper are as follows:

1. Considering the influence of multi-packet transmission, time-delay and packet dropout on the NCS system model, this control system is established as a discrete system model without time-delay by the switch system, which provides a modeling idea for the DCS with time-delay under the condition of multi-packet transmission.

2. The kernel function construction of multi-kernel LS-SVM is transformed into a comprehensive optimization problem of kernel function weight coefficient, and the coefficient is optimized by chaos adaptive artificial fish swarm algorithm, which provides a general solution for the support vector machine kernel function construction, reduces the dependence on prior knowledge, and greatly improves the performance.

3. Through LS-SVM and sliding window strategy, online compensation for packet dropout is realized, which provides a compensation strategy for time-delay and packet dropout in multi-packet transmission. The problem of time-delay and packet dropout can be solved effectively.

4. A novel PID approach law sliding-mode controller based on neural network is proposed. This controller can adjust PID parameters adaptively so as to realize performance on-line adjustment of sliding-mode controller, so that the controller can not only fit the rapidity of approach law, but also suppress chattering, as well as meet the control accuracy requirements.

The specific structure of this paper is as follows: the second part introduces the system modeling, sliding window strategy, multi-kernel LS-SVM packet dropout compensation strategy, the design of neural network PID approach law sliding-mode controller, while the third part describes the simulation results and the related analysis, and finally comes to the conclusion.

## Method

### Time-delay multi-packet transmission DCS modeling

The working environment of the engine is complex and the working conditions are harsh, there are inevitably uncertain factors such as parameter perturbation and external interference, so strong nonlinear characteristics are presented by the model. Aiming at the nonlinear model and the current aeroengine controller design, the commonly used method is to divide the flight envelope into several sub regions that meet certain performance indexes. In each region, a representative nominal operating point is selected, and a small deviation state space model of that point is established. The controller of the current envelope region is designed for the model. Therefore, the linear small deviation state space model is used to study the design of the controller [32–34]. It is assumed that DCS sensor and actuator are clock driven, controller is event driven. The data is time-stamped and transmitted in the form of multiple data packets without timing disorder. The control time-delay $\tau_{ca}$ and output time-delay $\tau_{se}$ are merged as $\tau$

($k$) according to reference [2]. On this basis, the discrete system model is established as:

$$\begin{cases} \boldsymbol{x}'(k+1) = \boldsymbol{A}'\boldsymbol{x}'(k) + \boldsymbol{B}'\boldsymbol{u}'(k - \tau(k)) \\ \boldsymbol{y}'(k) = \boldsymbol{C}'\boldsymbol{x}'(k) \end{cases} \tag{1}$$

where $\boldsymbol{x}'(k) \in \boldsymbol{R}^n$ is system state variable, $\boldsymbol{u}'(k) \in \boldsymbol{R}^m$ is control quantity input, $A', B', C'$ are dimensional coefficient matrixes. Suppose that $\tau(k)$ is a time-varying but bounded Markov random variable, the state space of $\tau(k)$ is $\Omega = \{0,1,2\}$, The time-delay state migration relation of the system is:

$$\begin{cases} \tau(k+1) = \tau(k) \cdot \pi_{i,j} \\ \displaystyle\sum_{j=0}^{2} \pi_{i,j} = 1 \qquad , i,j \in \boldsymbol{\Omega} \end{cases} \tag{2}$$

$\Pi = \boldsymbol{\pi}_{i,j}(i,j \in \Omega)$ is defined as the time-delay state migration matrix of the system. It is very difficult to design the sliding surface since there is the time-delay $\tau(k)$ in the aeroengine network control System Eq (1). Therefore, based on the predictive control idea [35,36], the original system is transformed into a time-delay free system by linear transformation. The linear transformation is defined as:

$$\boldsymbol{x}(k) = \boldsymbol{A}'^{\tau}\boldsymbol{x}'(k) + \sum_{i=0}^{\tau-1} \boldsymbol{A}'^{\tau-i-1}\boldsymbol{B}\boldsymbol{u}(k - \tau + i) \tag{3}$$

Substituting Eq (3) into System (1), the original system is converted as:

$$\boldsymbol{x}(k+1) = \boldsymbol{A}\boldsymbol{x}(k) + \boldsymbol{B}\boldsymbol{u}(k) \tag{4}$$

According to reference [37], System (4) is a state fully controllable system.

In Fig 1, if the state of the controlled object $\boldsymbol{x}(k)$ is divided into $m$ packets and transmitted to the controller, and the data received by the controller is $\bar{\boldsymbol{x}}(k)$, then:

$$x(k) = [x_1^{\mathrm{T}}(k), x_2^{\mathrm{T}}(k), \cdots, x_m^{\mathrm{T}}(k)]^{\mathrm{T}}; \; \bar{\boldsymbol{x}}(k) = [\bar{\boldsymbol{x}}_1^{\mathrm{T}}(k), \bar{\boldsymbol{x}}_2^{\mathrm{T}}(k), \cdots, \bar{\boldsymbol{x}}_m^{\mathrm{T}}(k)]^{\mathrm{T}}, \; \text{and}$$

$$\bar{\boldsymbol{x}}(k) = \boldsymbol{\Phi}_i \boldsymbol{x}(k) + \hat{\boldsymbol{\Phi}}_i \bar{\boldsymbol{x}}(k-1) \tag{5}$$

where, $\Phi_i = diag(0, \cdots 0, \varphi_{ii}, 0, \cdots 0)$, $\varphi_{ii} = 1$, $\hat{\boldsymbol{\Phi}}_i = \boldsymbol{I} - \boldsymbol{\Phi}_i$.

The input of LS-SVM packet dropout compensator is $\bar{\boldsymbol{x}}(k)$, which is the system control quantity and partial updated state quantity, and the output is $\tilde{\boldsymbol{x}}(k)$, which is the complete state quantity at current time. The relationship between LS-SVM compensation value $\tilde{\boldsymbol{x}}(k)$ and real value $\boldsymbol{x}(k)$ is as follows:

$$\tilde{\boldsymbol{x}}(k) = (1 + \boldsymbol{\sigma}(k))\boldsymbol{x}(k) \tag{6}$$

where, $\boldsymbol{\sigma}(k)$ is prediction error coefficient of LS-SVM compensation value and real value of system state.

During data transmission, in case of packet dropout, the non-updated data will be compensated and updated by the compensator. In this case, Eq (5) can be written as follows:

$$\bar{\boldsymbol{x}}(k) = \boldsymbol{\Phi}_i \boldsymbol{x}(k) + \hat{\boldsymbol{\Phi}}_i \tilde{\boldsymbol{x}}(k) \tag{7}$$

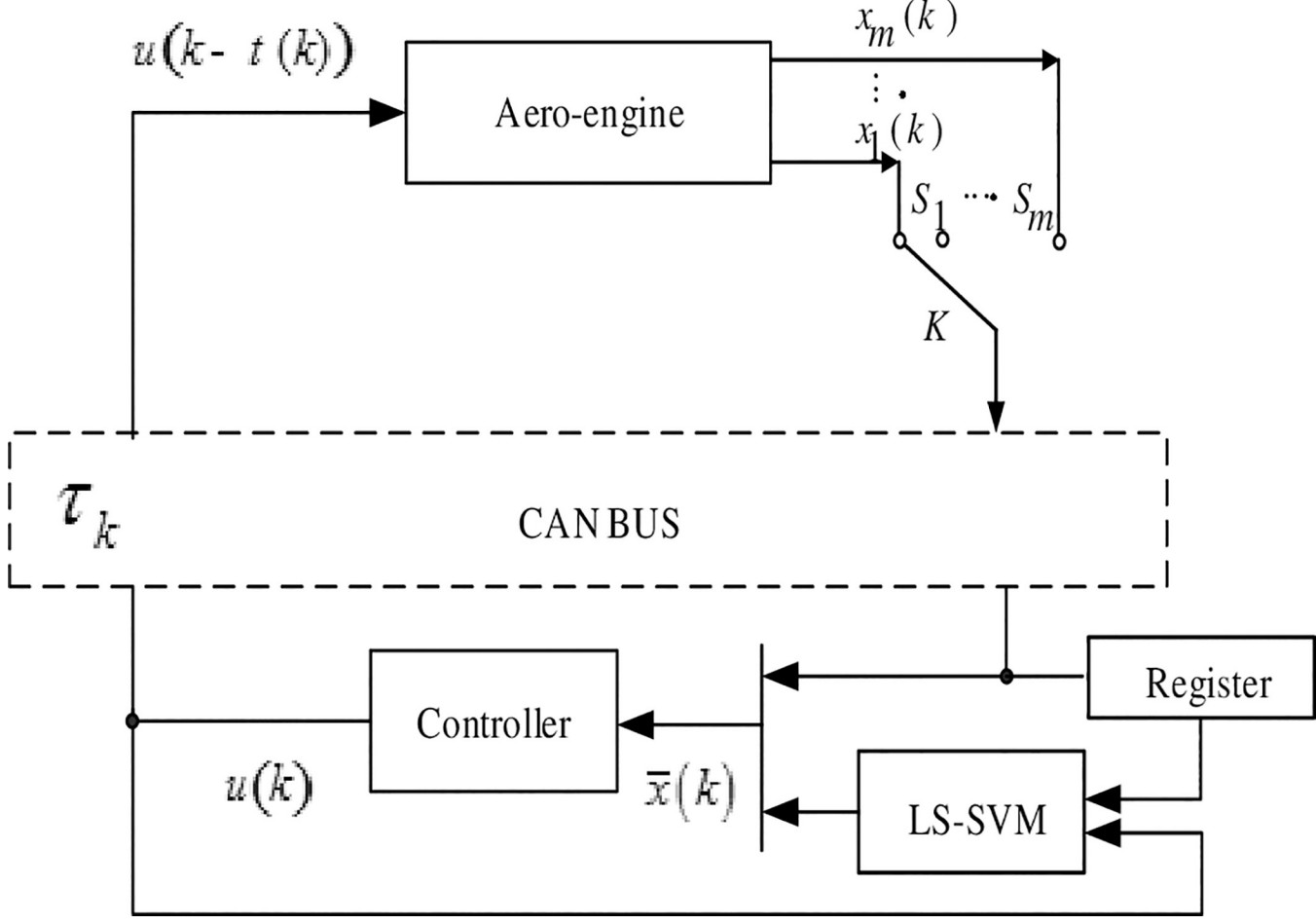

**Fig 1. Structure diagram of multiple packet transmission DCS for packet dropout compensator.**

In conclusion, the packet dropout compensation time-delay aeroengine DCS model is:

$$\begin{cases} \bar{\boldsymbol{x}}(k+1) = \boldsymbol{A}'\bar{\boldsymbol{x}}(k) + \boldsymbol{B}'\boldsymbol{u}(k) \\ \boldsymbol{y}(k) = \boldsymbol{C}'\boldsymbol{x}'(k) \end{cases} \tag{8}$$

## Online compensation of sliding time window multi-Kernel LS-SVM

LS-SVM solves the problem that the amount of regression calculation increases with the number of samples in traditional SVM learning algorithm [38]. Given the sample sequence $(x_1,y_1)$, $(x_2,y_2), \cdots, (x_i,y_i), \cdots, (x_i,y_i)$, assuming that $\boldsymbol{x}_i \in \boldsymbol{R}^n$ represents the input vector and $\mathbf{y}_i \in \boldsymbol{R}$ represents the output vector, the LS-SVM solution problem can be expressed as:

$$\min \quad \phi(\boldsymbol{w}, \boldsymbol{b}, \boldsymbol{e}) = \frac{1}{2}\boldsymbol{w}^{\mathrm{T}}\boldsymbol{w} + \frac{\gamma}{2}\sum_{i=1}^{l}\boldsymbol{e}_i^2$$

$$s.t. \quad \begin{cases} \boldsymbol{y}_i = \boldsymbol{w}^{\mathrm{T}}\varphi(\boldsymbol{x}_i) + \boldsymbol{b} + \boldsymbol{e}_i \\ \\ i = 1, 2, \ldots, l \end{cases} \tag{9}$$

where $\varphi(\cdot): \boldsymbol{R}^n \rightarrow \boldsymbol{R}^{nh}$ is mapping function; $\boldsymbol{w} \in \boldsymbol{R}^{nh}$ is weight coefficient; $\boldsymbol{e}_i \in \boldsymbol{R}$ is error vector;

$b \in \mathbf{R}$ is bias coefficient; $\gamma > 0$ is penalty factor, from Eq (9), we can see that the penalty factor has a great influence on the prediction error. According to the theory of LS-SVM, the basic equations are:

$$\begin{bmatrix} 0 & \mathbf{I'}^T \\ \mathbf{I'} & \mathbf{\Omega} + \gamma^{-1}\mathbf{I} \end{bmatrix} \begin{bmatrix} \mathbf{b} \\ \mathbf{\alpha} \end{bmatrix} = \begin{bmatrix} 0 \\ \hat{\mathbf{y}} \end{bmatrix} \tag{10}$$

where $\hat{\mathbf{y}} = [y_1, y_2, \ldots, y_l]_{l \times 1}$; $\mathbf{I'} = [1,1,\ldots,1]_{l \times 1}$; $\mathbf{\alpha} = [\alpha_1, \alpha_2, \ldots, \alpha_l]_{l \times 1}^T$; $\mathbf{I}$ is identity matrix; $\Omega = \Omega_{l \times l}$; $\Omega_{i,j} = \mathbf{K}(\mathbf{x}_i, \mathbf{x}_j) = \varphi(\mathbf{x}_i)\varphi(\mathbf{x}_j)$, $i,j = 1,2,\ldots,l$; $\mathbf{K}(\mathbf{x}_i, \mathbf{x}_j)$ is kernel function.

According to the principle of SVM, the selection of kernel function has a significant impact on the final regression prediction. Kernel function can greatly reduce the computational complexity for determined feature space and corresponding mapping [39]. There are three common kernel functions.

1. Polynomial Kernel Function:

$$\mathbf{K}_\mathrm{p}(\mathbf{x}, \mathbf{y}) = [\lambda(\mathbf{x}^\mathrm{T}\mathbf{y}) + c]^\mathrm{d} \tag{11}$$

2. Sigmoid Kernel Function:

$$\mathbf{K}_\mathrm{s}(\mathbf{x}, \mathbf{y}) = \tanh(\eta\mathbf{x}^\mathrm{T}\mathbf{y} + k_2) \tag{12}$$

3. Gauss Kernel Function:

$$\mathbf{K}_\mathrm{g}(\mathbf{x}, \mathbf{y}) = \exp(-\frac{\|\mathbf{x} - \mathbf{y}\|^2}{2\sigma^2}) \tag{13}$$

where $\mathbf{x}, \mathbf{y}$ is input space vector, $\lambda, c, d, \eta, \sigma$ are parameters of kernel function.

In addition to the above three common kernel functions, there are multiple quadric surface kernel function, orthogonal polynomial expansion kernel function, Fourier expansion kernel function and various improved kernel functions. For the convenience of explanation, this paper only takes the common kernel functions mentioned above as examples.

There are several properties as for the kernel functions [40]:

1. Assuming that both $K_1$ and $K_2$ are kernel functions, $\alpha_1$ and $\alpha_2$ are both positive real numbers, then $K = \alpha_1 K_1 + \alpha_2 K_2$ shall be a kernel function also;

2. Assuming that both $K_1$ and $K_2$ are kernel functions, then $K = K_1 \cdot K_2$ shall be a kernel function as well;

3. Assuming that $K_1$ is a kernel function, then $K = \exp(K_1)$ shall be a kernel function also;

According to the above three properties, numerous different kernel functions can be obtained, and their combination relationship is shown in Fig 2.

$$\mathbf{K} = \sum_{i=1}^{n} \omega_i \mathbf{K}_i \tag{14}$$

where $\mathbf{K}_i$ is a new kernel function obtained by any permutation and combination of kernel functions with the above three properties; $\omega_i$ is weight coefficient of each combined kernel function. The multi-kernel function synthesizes the characteristics of various kernel functions,

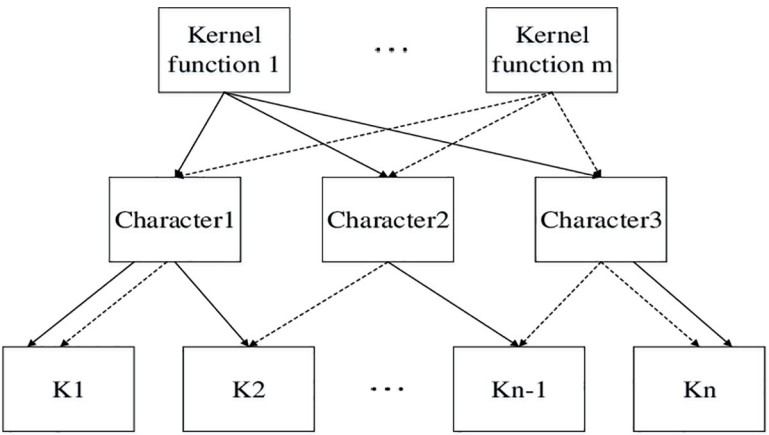

**Fig 2. Combination diagram of Kernel function.**

and adjusts the influence of different kernel functions on the prediction accuracy by the size of the weight coefficient, so as to transform the selection of kernel functions into the optimization solution of the kernel functions weight, further to synthesize the characteristics of each kernel function and improve the accuracy of SVM. The corresponding objective function is:

$$
\textbf{\textit{Fitness}} = \left| 1 - \frac{(l\sum_{i=1}^{l} \hat{\boldsymbol{y}}_i \boldsymbol{y}_i - \sum_{i=1}^{l} \hat{\boldsymbol{y}}_i \sum_{i=1}^{l} \boldsymbol{y}_i)^2}{(l\sum_{i=1}^{l} \hat{\boldsymbol{y}}_i^2 - (\sum_{i=1}^{l} \hat{\boldsymbol{y}}_i)^2)(l\sum_{i=1}^{l} \boldsymbol{y}_i^2 - (\sum_{i=1}^{l} \boldsymbol{y}_i)^2)} \right| \tag{15}
$$

The constraints are:

$$
\begin{cases} \sum_{i=1}^{n} \omega_i = 1 (i = 1, 2, \cdots, n) \\ \omega_i > 0 (i = 1, 2, \cdots, n) \end{cases} \tag{16}
$$

The closer the fitness function is to 0, the higher the prediction accuracy is. Where, $l$ represents the number of test samples, $\hat{y}_i$ represents the true value of the sample, and $y_i$ is the estimated value of the sample.

The coefficients $\alpha$ and $b$ can be obtained by solving Eq (10), so as to get the least squares support vector regression model:

$$
\boldsymbol{y}(x) = \sum_{i=1}^{l} \boldsymbol{\alpha}_i K(x, x_i) + \boldsymbol{b} \tag{17}
$$

The characteristic matrix is defined as: $\boldsymbol{Q} = \Omega + \gamma^{-1}\boldsymbol{I}$, where:

$$
\begin{cases} \boldsymbol{b} = \dfrac{\boldsymbol{I}^T \boldsymbol{Q}^{-1} \hat{\boldsymbol{y}}}{\boldsymbol{I}^T \boldsymbol{Q}^{-1} \boldsymbol{I}} \\ \boldsymbol{\alpha} = \boldsymbol{Q}^{-1}(\hat{\boldsymbol{y}} - \boldsymbol{I}' \times \boldsymbol{b}) \end{cases} \tag{18}
$$

In order to realize online packet dropout compensation, this paper uses sliding time window strategy and LS-SVM to model online prediction.

The sliding time window strategy updates the training data every time the time window moves. Assuming that the length of the time window is $L$, the value of the length is related to the number of samples.

According to the theory of least square support vector machine, the key to solve the regression model is to find the inverse matrix of $\boldsymbol{Q}$, to make $\boldsymbol{Q}_L = \Omega_L + (1/\gamma)\boldsymbol{I}$, where $\boldsymbol{Q}_L \in \boldsymbol{R}^{L \times L}$; $\Omega_L \in \boldsymbol{R}^{L \times L}$; $\Omega_{i,j} = \boldsymbol{K}(\boldsymbol{x}_i, \boldsymbol{x}_j)$ represents kernel function, $i,j = 1,2,\ldots,L$. So, the sample updating problem is equivalent to the updating of $\boldsymbol{Q}_L^{-1}$. Specific update steps of packet dropout online compensation LS-SVM algorithm of sliding time window network control system can be referred to [41].

## Design of neural network sliding mode controller

**Theorem** If the expression of PID approach law is given as:

$$\dot{\boldsymbol{s}} = -l(\boldsymbol{s} + \mathrm{sgn}(\boldsymbol{s})l) - \mathrm{sgn}(\boldsymbol{s})m \int_{t_0}^{t} |\boldsymbol{s}|dt - n\dot{\boldsymbol{s}} \tag{19}$$

where $l>0$ is proportionality coefficient, $m>0$ is integral coefficient, $n>0$ is differential coefficient. $t_0$ is the time of system reaching sliding surface for the first time, $t$ represents the current time. The sliding surface of discrete sliding mode control is designed as:

$$\boldsymbol{s}(k) = \boldsymbol{F} \cdot \bar{\boldsymbol{x}}(k) = \begin{bmatrix} z_1 & 0 \\ 0 & 1 \end{bmatrix} \begin{bmatrix} \bar{\boldsymbol{x}}_1(k) \\ \bar{\boldsymbol{x}}_2(k) \end{bmatrix} \tag{20}$$

where the sliding surface constant matrix $\boldsymbol{F}$ is:

$$\boldsymbol{F} = diag[z_1 \ 1] \tag{21}$$

Then the PID approach law satisfies the conditions of existence and arrival of the sliding mode, and the sliding-mode controller is asymptotically stable, the control quantity $\boldsymbol{u}(k)$ shall be:

$$\boldsymbol{u}(k) = -(\boldsymbol{F}B)^{-1}[\boldsymbol{F}A'\bar{x}(k) - \boldsymbol{s}(k+1)] \tag{22}$$

Proof:
When $\boldsymbol{s}>0$ and $\boldsymbol{s} \rightarrow 0^+$, there exist:

$$\lim_{s \rightarrow 0^+} \dot{\boldsymbol{s}} = -\frac{l}{(1+n)}(\boldsymbol{s} + \mathrm{sgn}(\boldsymbol{s})l)$$
$$-\mathrm{sgn}(\boldsymbol{s})\frac{m}{(1+n)} \int_{t_0}^{t} |\boldsymbol{s}|dt < 0 \tag{23}$$

Thus $\boldsymbol{s}\dot{\boldsymbol{s}} < 0$ is satisfied. For the same reason, when $\boldsymbol{s}<0$ and $\boldsymbol{s} \rightarrow 0^-$,

$$\lim_{s \rightarrow 0^-} \dot{\boldsymbol{s}} = -\frac{l}{(1+n)}(\boldsymbol{s} + \mathrm{sgn}(\boldsymbol{s})l)$$
$$-\mathrm{sgn}(\boldsymbol{s})\frac{m}{(1+n)} \int_{t_0}^{t} |\boldsymbol{s}|dt > 0 \tag{24}$$

$\boldsymbol{s}\dot{\boldsymbol{s}} < 0$ is satisfied as well.

Based on the above analysis, the proposed PID approach law satisfies the conditions of existence and arrival of the sliding mode.

When the system does not reach the sliding surface, the effect of the integral term is 0. When $s(t) = 0$, the time of system reaching sliding surface for the first time can be solved by Eqs (23) and (24):

$$\begin{cases} t_0 = \dfrac{1+n}{l}\ln\dfrac{s(0)+l}{l}, s > 0 \\[2mm] t_0 = \dfrac{1+n}{l}\ln\dfrac{s(0)+l}{l}, s \leq 0 \end{cases} \tag{25}$$

According to Eq (25), the arrival time $t_0$ is finite value.

According to the compensation modeling of the aeroengine network control system, the corresponding state space model is shown in Eq (8), assuming the number of state variables is 2, the sliding surface of discrete sliding mode control is designed as:

$$s(k) = \mathbf{F} \cdot \bar{\mathbf{x}}(k) = \begin{bmatrix} z_1 & 0 \\ 0 & 1 \end{bmatrix} \begin{bmatrix} \bar{\mathbf{x}}_1(k) \\ \bar{\mathbf{x}}_2(k) \end{bmatrix} \tag{26}$$

where $\mathbf{F}$ is the sliding surface constant matrix. Then Eq (6) is equivalent to:

$$\begin{cases} \bar{\mathbf{x}}_1(k+1) = A'_{11}\bar{\mathbf{x}}_1(k) + A'_{12}\bar{\mathbf{x}}_2(k) + \mathbf{B}_1\mathbf{u}(k) \\ \bar{\mathbf{x}}_2(k+1) = A'_{21}\bar{\mathbf{x}}_1(k) + A'_{22}\bar{\mathbf{x}}_2(k) + \mathbf{B}_2\mathbf{u}(k) \end{cases} \tag{27}$$

$$s(k) = z_1\bar{\mathbf{x}}_1(k) + \bar{\mathbf{x}}_2(k) \tag{28}$$

When reaching sliding surface for the first time, it is known that the following conditions are met:

$$s(k_0) = 0, k_0 \neq 0 \tag{29}$$

Then simultaneously solve Eqs (27) and (28):

$$\begin{cases} \bar{\mathbf{x}}_1(k+1) = (A'_{11} - A'_{12}z_1)\bar{\mathbf{x}}_1(k) + \mathbf{B}_1\mathbf{u}(k) \\ \bar{\mathbf{x}}_2(k+1) = A'_{21}\bar{\mathbf{x}}_1(k) + A'_{22}z_1\bar{\mathbf{x}}_1(k) + \mathbf{B}_2\mathbf{u}(k) \end{cases} \tag{30}$$

After $k$ is determined, the sliding surface constant matrix $\mathbf{F}$ can be solved:

$$\mathbf{F} = diag[z_1 \quad 1] \tag{31}$$

Discrete sliding mode control is a kind of quasi sliding mode motion. It is difficult for the system to stabilize on the sliding surface. The moving point of the system moves back and forth in the boundary layer on both sides of the sliding surface, thus forming chattering. According to the analysis of continuous sliding-mode PID approach law, for discrete sliding-mode control, the arrival condition equivalent to the condition $s\dot{s} < 0$ is:

$$[s(k+1) - s(k)]s(k) < 0 \tag{32}$$

However, it can be seen from reference [15] that Eq (32) is only a necessary condition for the existence of discrete quasi sliding mode motion, but not a sufficient condition. To solve

this problem, Sarpturk proposes a sufficient condition for discrete sliding mode arrival:

$$|s(k+1)| < |s(k)| \tag{33}$$

According to the analysis of continuous approach law, the discrete sliding surface function can be expressed as:

$$
\begin{cases}
s(k+1) = \dfrac{-lT+n+1}{n+1}s(k) - \dfrac{mT}{n+1}\displaystyle\sum_{k_0}^{k}Ts(k) - \dfrac{l^2T}{n+1}, s(k), 0 \\[4mm]
s(k+1) = \dfrac{-lT+n+1}{n+1}s(k) + \dfrac{mT}{n+1}\displaystyle\sum_{k_0}^{k}Ts(k) + \dfrac{l^2T}{n+1}, s(k) \le 0
\end{cases}
\tag{34}
$$

According to Eq (34), at this time, no matter $s(k) > 0$ or $s(k) \le 0$, can meet the requirements of Eq (33). Furthermore, the stability of the PID approach law sliding-mode controller is analyzed, and the Lyapunov function is defined:

$$V(k) = s^2(k) \tag{35}$$

Thus:

$$\Delta V(k) = s^2(k+1) - s^2(k) \tag{36}$$

Since Eq (35) is satisfied, $\Delta V(k) < 0$, which can prove that the sliding-mode controller is asymptotically stable. Thus, the equivalent control quantity $u(k)$ is shown as Eq (22).

Whether the PID approach law can keep small chattering when the control speed is ensured, it depends on three parameters: proportion, integral and differential. In order to achieve efficient sliding-mode control, these parameters should be adjusted adaptively according to the time of reaching the sliding surface. Therefore, considering the strong nonlinear mapping ability of neural network [42], a sliding-mode controller of PID approach law parameters online adjustment based on neural network is proposed.

The input of the neural network is the sliding mode switching function $s(k)$ and its variation $\Delta s(k)$, where $\Delta s(k) = s(k+1) - s(k)$. These two inputs can reflect the current state of the sliding surface and the future movement trend. The outputs are three parameters of the PID approach law: $l,m,n$. Radial basis neural network belongs to the multilayer feedforward neural network with strong nonlinear mapping ability [43].

In this paper, the generalized Radial Basis Function (RBF) network is applied, and its structure diagram is shown in Fig 3.

The specific calculation of the generalized RBF nonlinear mapping is based on the method in [44], which will not be discussed here.

## Results and discussions

Structure of aero-engine DCS semi-physical platform is shown in Figs 4 and 5. It is composed of five parts, the model computer, the control computer, the intelligent sensor, the intelligent actuator and the CAN bus. The aero-engine model runs on the model computer, and the analog signal is transformed into corresponding digital signal by the intelligent sensor. The control computer receives digital signals from CAN bus, and then the control algorithm is operated to output control signal transferred to CAN bus. Real-time display of engine operation data and curve, controller parameter adjustment, fault simulation, and communication detection can be realized in the control panel. The intelligent actuator can receive control

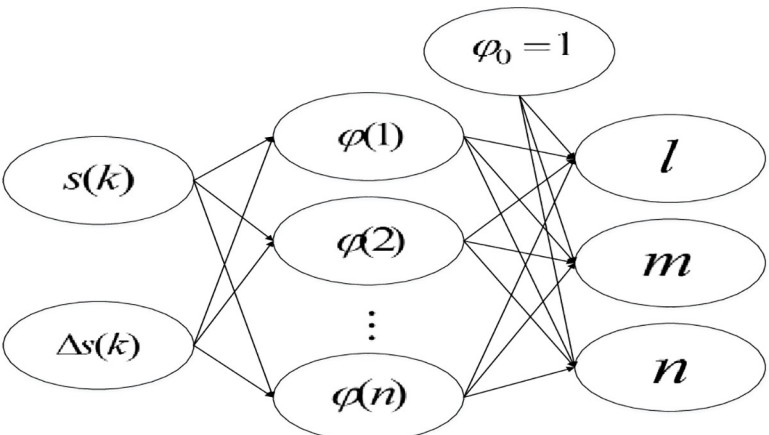

**Fig 3. General RBF network structure.**

signal from CAN bus, output oil supply signal or geometric channel signal and transfer that to the model computer for speed control.

In this method, the controller is designed for the multi packet transmission network control system. Therefore, the effect of packet dropout compensation is mainly determined by the prediction relative error of the actual data and the data at relevant packet dropout rate. The smaller the error is, the better the compensation effect is. For the sliding mode controllers with different approach laws, the response time and steady-state error are mainly considered to determine the quality of the controller. With shorter response time, better control speed and smaller steady-state error, the better chattering suppression effect and the higher precision of the sliding mode controller can be obtained.

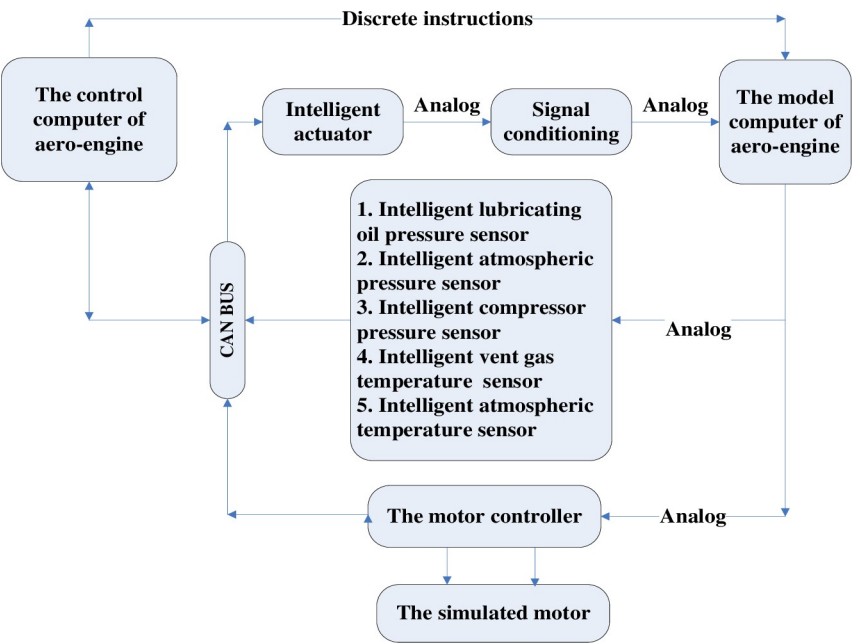

**Fig 4. The structure configuration of aero engine DCS semi-physical platform.**

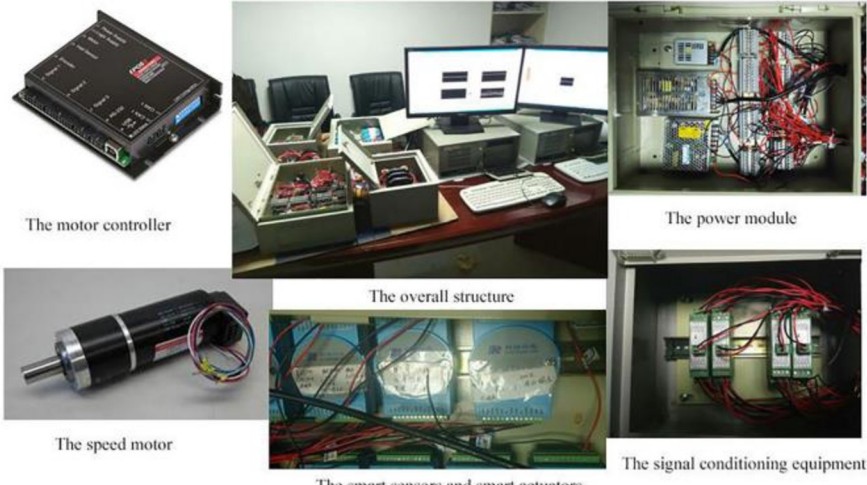

**Fig 5. The configuration of aero engine DCS semi-physical platform.**

It is defined that the sampling period of the twin rotor turboshaft engine network control system is 20ms, and assuming that the system condition is: $H = 0$km, $Ma = 0$, the engine speed is $n_H = 100\%$. Then the parameter matrix of system state space model is:

$$A' = \begin{bmatrix} -0.8641 & 0.1491 & -0.01559 \\ -0.0073 & 0.9445 & -0.00532 \\ 0.4759 & -0.08775 & 0.5990 \end{bmatrix}, \; B' = \begin{bmatrix} 0.01935 & 0.00468 \\ 0.01731 & 0.01059 \\ 0.1853 & -0.0959 \end{bmatrix}, \; \boldsymbol{x} = \begin{bmatrix} n_L & n_H & p_3 \end{bmatrix}^T, \; \boldsymbol{u} = \begin{bmatrix} m_f & A_8 \end{bmatrix}^T$$

where $n_L$ is low pressure rotor speed, $n_H$ is high pressure rotor speed, $p_3$ is air-compressor outlet total pressure, $m_f$ is main fuel flow, $A_8$ is the critical cross-sectional area of tail nozzle.

Time-delay state migration matrix is defined as:

$$\boldsymbol{\Pi}_1 = \begin{bmatrix} 0.3 & 0.4 & 0.3 \\ 0.3 & 0.2 & 0.5 \\ 0.1 & 0.6 & 0.3 \end{bmatrix} \tag{37}$$

Fig 6 shows the time-delay distribution of $\Pi_1$:

In order to reduce the calculation cost, the number of corresponding sub kernel functions is $n = 6$, the composition is shown in Table 1.

Training with packets that are not lost, the kernel function parameters and structure parameters of multi-kernel LS-SVM are optimized by the chaos adaptive artificial fish swarm method in reference [2], assuming that the number of artificial fish $NUM = 30$, the maximum iterations $Iterate\_times = 170$, the initialize field of view $Visual = 15$, the crowding factor $\varphi = 0.4$, the foraging attempts number $Try\_number = 10$, the attenuation factor $\alpha = 0.4$, $\beta = 0.3$, the threshold $\delta = 0.5$, the optimization results are shown in Fig 7.

The weight optimization curve of each corresponding kernel function is shown in Fig 8. Similarly, other structural parameters of LS-SVM can be obtained.

The initial parameters value of the sliding-mode controller is set as proportional coefficient $l = 30$, integral coefficient $m = 1$, differential coefficient $n = 5$, and the number of layers of neural network is set as 8, the number of neurons in the hidden layer is 4, the corresponding

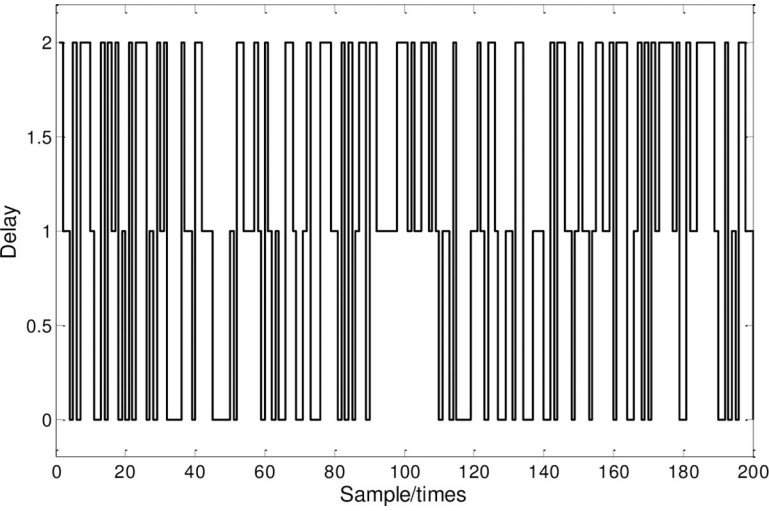

**Fig 6. Delay distribution.**

weight coefficient is obtained from the training samples, and the sliding surface constant matrix can be further calculated by pole assignment:

$$\boldsymbol{F} = diag[\, z_1 \quad 1 \quad 1 \,] = diag[\, 4.27 \quad 1 \quad 1 \,] \tag{38}$$

The control quantity can be calculated according to Eq (22). Firstly, the online prediction compensation of sliding time window multi-kernel LS-SVM is verified. The speed change curve of given high-pressure rotor is shown as the red curve in Fig 9, the combination kernel function LS-SVM [19] based on sliding time window strategy and the optimized multi-kernel LS-SVM are used for the packet dropout prediction compensation under 30% and 60% packet loss rate respectively. The prediction comparison results are shown in Fig 9. The corresponding prediction relative error comparison is shown in Table 2.

As can be seen from the predicted compensation results in Fig 9 and Table 2, when the packet loss rate is 30% and 60%, the average error of packet dropout prediction of multi-kernel LS-SVM reduces 29.21% and 44.66% compared with that of combined kernel LS-SVM, and when the packet loss rate is small, the change situation of state quantity that without packet dropout can be reproduced basically. It shows that the prediction and compensation accuracy of multi-kernel LS-SVM is higher than that of combined kernel LS-SVM regardless of packet loss rate.

Furthermore, the influence of online compensation on neural network sliding-mode control under different packet loss rate is considered. In Fig 10, the proposed multi-kernel

**Table 1. Composition of Kernel functions.**

| Kernel Function | Expression |
|:---:|:---:|
| $K_1$ | $K_p + K_g$ |
| $K_2$ | $K_p \cdot K_g \cdot K_s$ |
| $K_3$ | $K_s \cdot K_g$ |
| $K_4$ | $\exp(K_s + K_g)$ |
| $K_5$ | $\exp(K_s \cdot K_g \cdot K_p)$ |
| $K_6$ | $K_p \cdot \exp(K_s)$ |

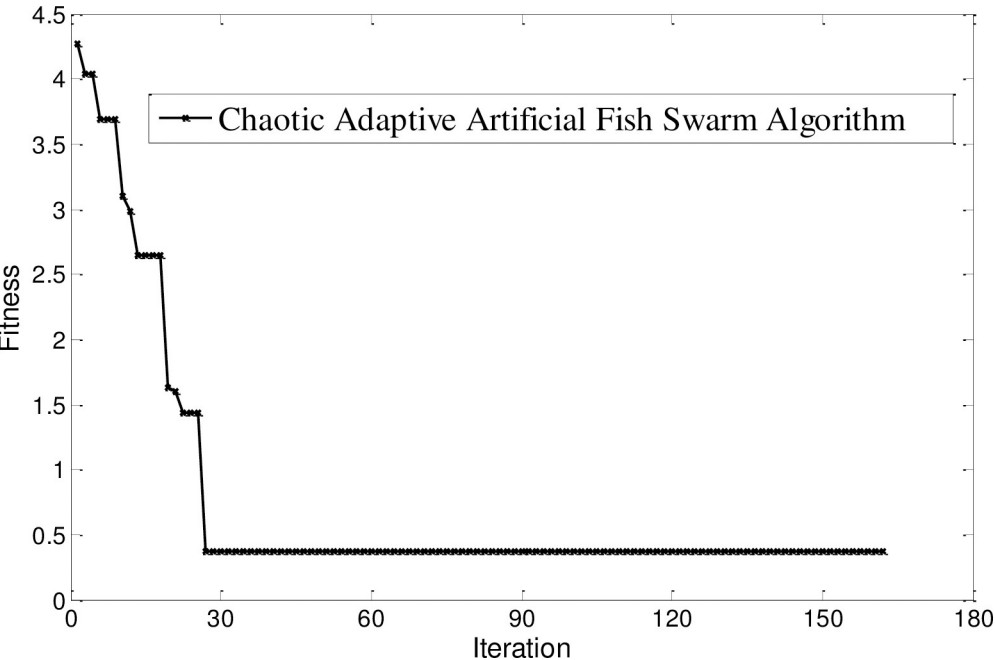

**Fig 7. Optimization results.**

function optimized LSSVM compensation method is compared with PSO neural network compensation, Gauss kernel function LS-SVM compensation, combined kernel function LS-SVM compensation and uncompensated method under the condition of RBF-PID approach law sliding mode controller. In Fig 11, the packet loss rate is sixty percent. Under the condition of RBF-PID approach law sliding mode controller, the proposed multi-kernel function optimization LSSVM compensation method is compared with PSO neural network

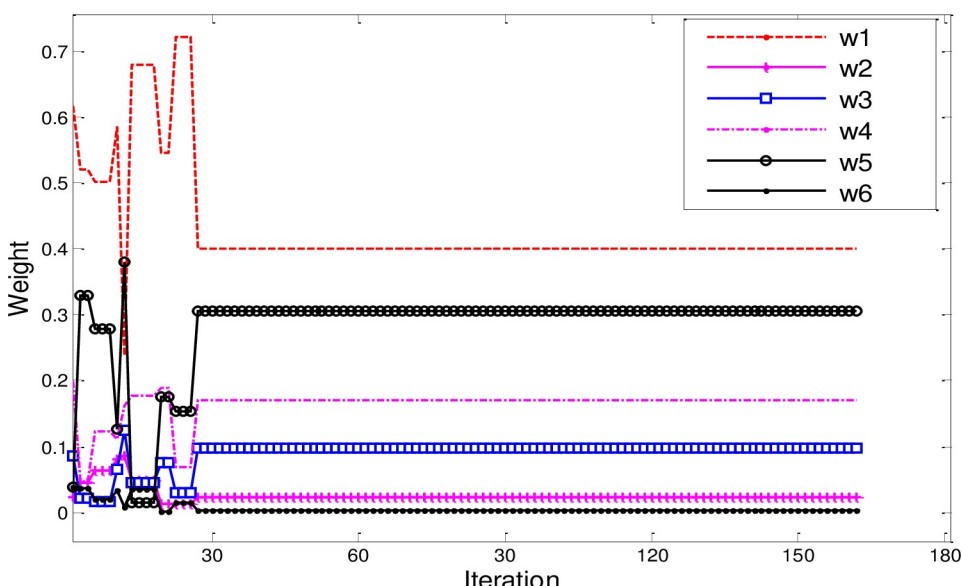

**Fig 8. Weight optimization results of Kernel functions.**

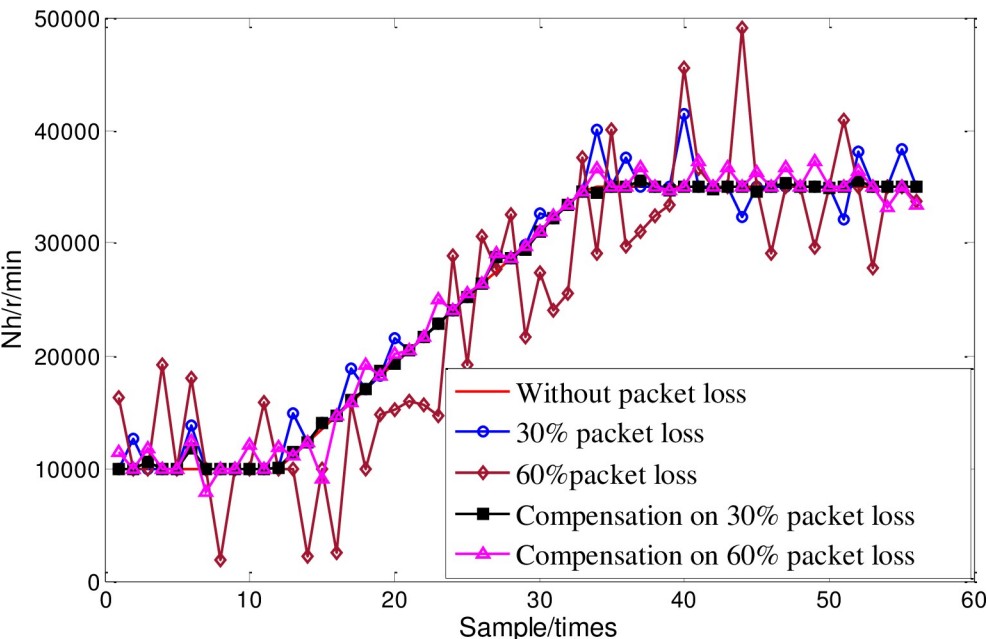

**Fig 9. Packets loss prediction compensation comparison.**

compensation, Gauss kernel function LS-SVM compensation, combined kernel function LS-SVM compensation and uncompensated method. It can be seen from the figure that no matter the packet loss rate is 30% or 60%, the speed and steady-state performance of the neural network sliding-mode control response based on the optimized multi-kernel LS-SVM online compensation is better than other methods, which further proves that the neural network sliding-mode control effect under the packet dropout condition can be improved by the data packet online prediction compensation, and can achieve better control effect under certain packet loss rate.

In this paper, the comparative law of approach, which includes the fixed parameter PID approach law, the fuzzy exponential approach law, the segment approach law, the exponential approach law, and the global approach law are selected. These laws are all traditional laws of approach. The exponential law of approach is famous for its fast response speed. The piecewise law of approach is achieved by considering the system performance in different time periods and applying different characteristics of the law of approach in a certain performance. The fuzzy exponential approach law realizes superior sliding mode control through adjusting relevant parameters on-line by fuzzy theory, and the global approach law realizes control by considering global characteristics. All of the above approaches have been proved to be effective and widely used. The fixed parameter PID approach law mainly compares the advantages of RBF-PID adaptive adjustment.

**Table 2. Comparison of different compensation condition.**

| Condition | Prediction relative error |
|---|---|
| 30% packet loss | 32.45% |
| Compensation on 30% packet loss | 3.24% |
| 60% packet loss | 56.52% |
| Compensation on 60% packet loss | 11.86% |

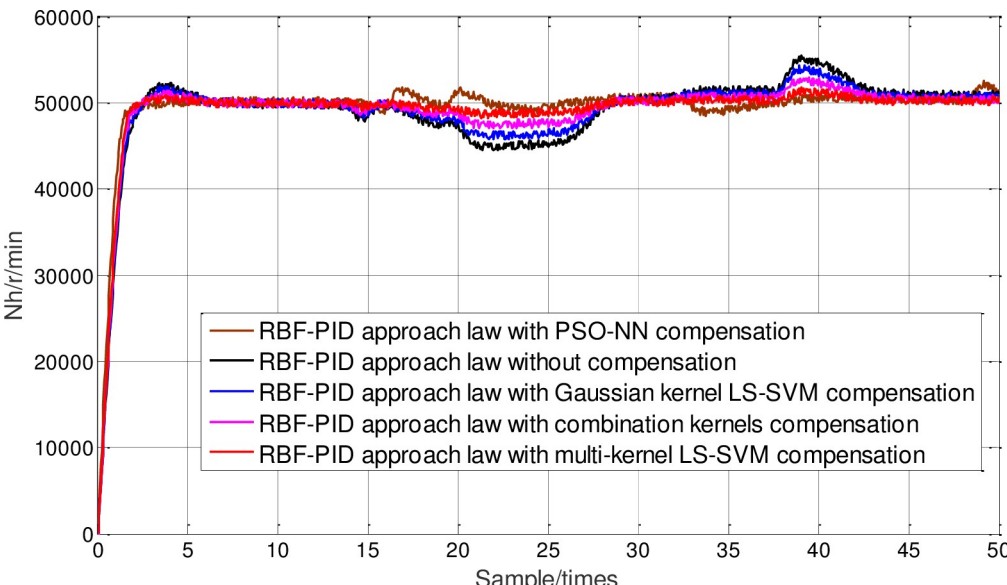

**Fig 10. Controlling comparison under 30% packets loss rate.**

In order to prove the superiority of neural network PID approach law sliding-mode control, under the condition of packet loss rate at 20%, the given reference tracking signal is a step signal with high-pressure speed equal to 50000r/min, and under the condition of that multi-kernel LS-SVM online packet dropout compensation optimized by sliding time window, the fixed parameter PID approach law, the fuzzy exponential approach law, the segment approach law, the exponential approach law, and the global approach law and neural network PID approach law sliding-mode control are respectively used to control the distributed system of aeroengine.

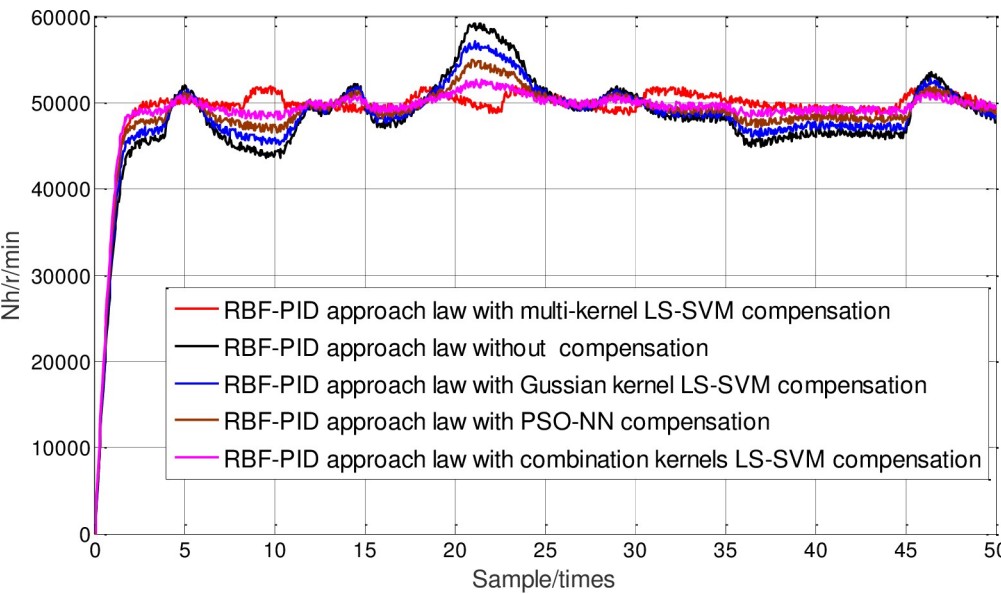

**Fig 11. Controlling comparison under 60% packets loss rate.**

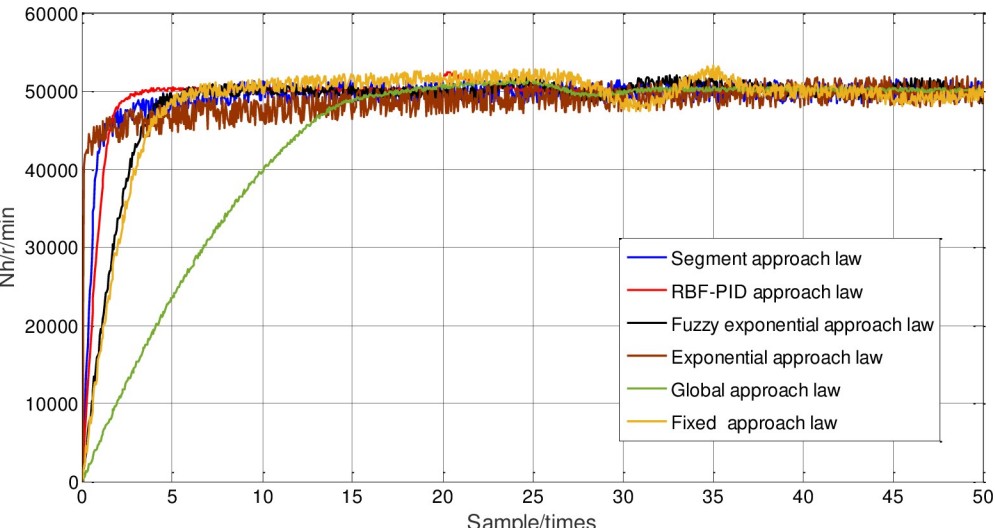

**Fig 12. Comparison of high pressure speed response with different approach laws.**

The high-pressure speed response curve is shown in Fig 12, and the corresponding steady-state amplification diagram under high-pressure speed is shown in Fig 13.

The specific steady-state chattering results after 100 sampling periods are shown in Table 3.

It can be seen that although the response speed of the RBF-PID approach law is the fastest, its chattering amplitude value is significantly greater than other methods. Compared with the piecewise approach law, the chattering of the fuzzy power approach law is greatly reduced, but its response regulation time is significantly increased. Compared with the fixed parameter PID approach law, the fuzzy exponential approach law, the segment approach law, the exponential approach law, and the global approach law, the steady-state error of neural network PID approach law sliding-mode control is reduced by 7.86%,2.13%,5.5%,10.07% and 0.49% respectively, which shows that the chattering reduction has been greatly improved, and the response curve can quickly rise to the target value and keep a small steady-state error.

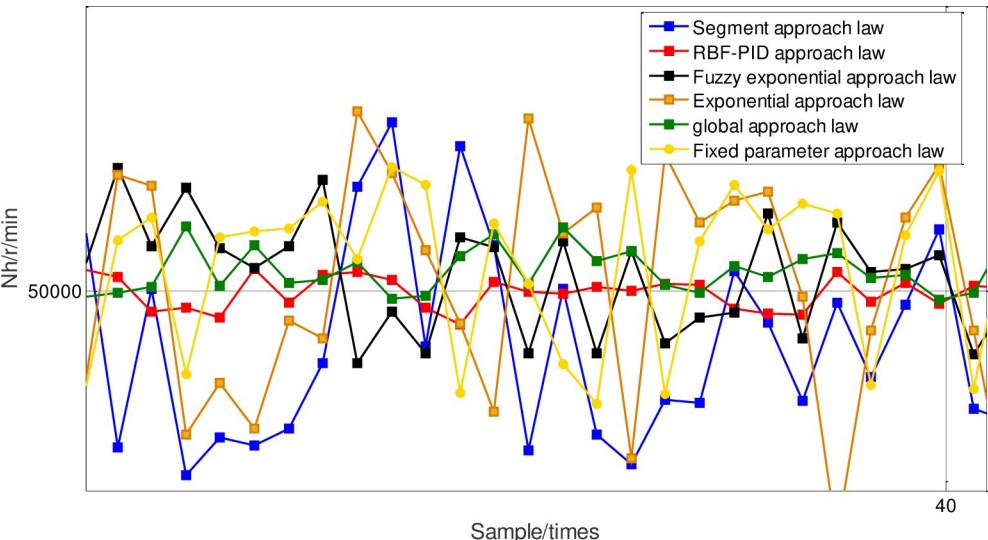

**Fig 13. Amplification of high pressure speed response with different approach laws.**

**Table 3. Comparison of high pressure speed steady-state chattering results.**

| Approach law | Response time(s) | Steady-state error(%) |
|---|---|---|
| Fixed parameter PID | 5.2 | 8.59 |
| Exponential | 11.3 | 10.8 |
| Global | 14.8 | 1.22 |
| Segment | 4.8 | 6.23 |
| Fuzzy exponential | 5.1 | 2.86 |
| RBF-PID | 3.9 | 0.73 |

Then, the chattering under different approach laws is analyzed by the response curve of the control quantity $u(k)$, as shown in Figs 14 and 15, it can be seen that the chattering amplitude of the control quantity $u(k)$ under the neural network PID approach law sliding-mode control, is significantly smaller than that under PID approach law. Table 4 shows the average steady-state errors of different approach laws after 100 sampling periods. The average steady-state errors of the neural network PID approach law sliding-mode control are significantly smaller than those of other methods. From the point of view of steady-state errors, it further shows that the chattering of neural network PID approach law is much weaker.

Reason Analysis: Because the piecewise approach law realizes the switching between the two approaches through distance from the sliding surface, in the initial stage of response, the approach speed is mainly considered, so the response speed is faster. However, after the approach law is switched, the chattering reduction is mainly considered, so the response curve will have an obvious turning point. However, the state variable has not reached the sliding surface at this time, so after the switch, the approach law does not reduce chattering, but slows down the response speed. The choice of switching time in this method will have a great influence on the final control effect. The fuzzy power approach law can adjust the speed of the approach law online. Its design goal is mainly to reduce the chattering of the system, enhance the robustness of the system to external interference and parameter perturbation, so its robustness is strong, but the response speed is slow. The neural network PID approach law can make the proportion, integral and differential parameters adjustable through the nonlinear mapping

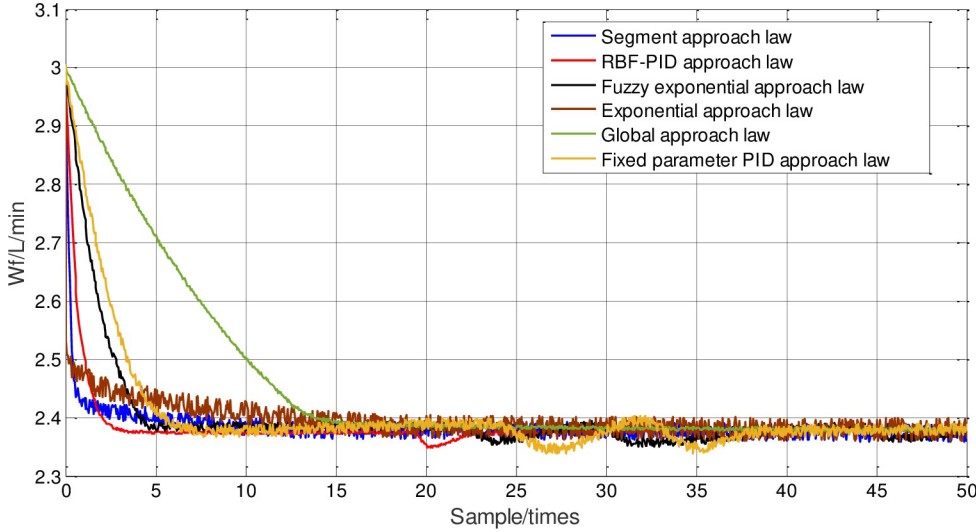

**Fig 14. Comparison of fuel supply response with different approach laws.**

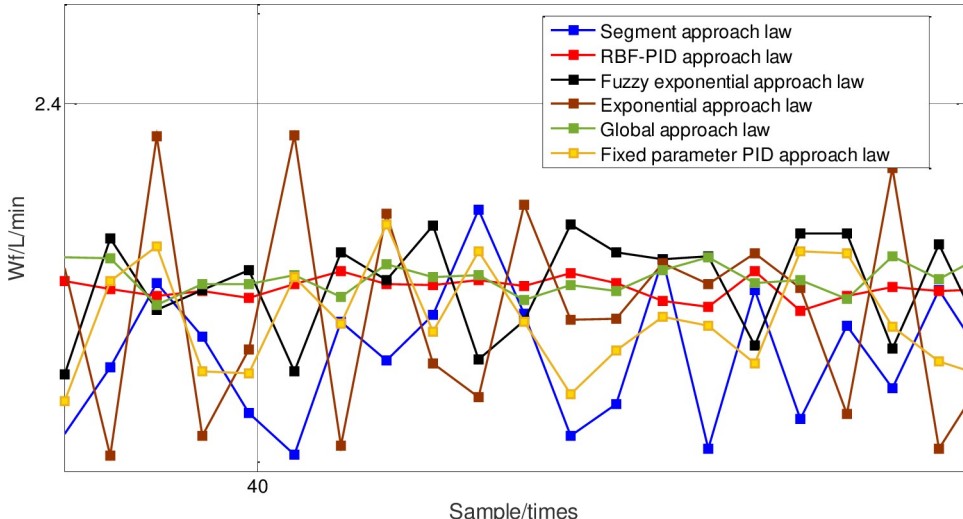

**Fig 15. Magnification of fuel supply with different approach laws.**

ability of neural network. It can speed up the approach speed by increasing the proportion coefficient in the early stage, and when reaching the sliding surface in the later stage, it can reduce the proportion coefficient, increase the integral coefficient to reduce the chattering amplitude, decrease the steady-state error, and increase the differential coefficient to suppress chattering, which takes into account the response speed and the chattering suppression at the same time.

## Conclusion

The following conclusions are drawn in this paper:

1. In this paper, the combined kernel function construction of multi-kernel support vector regression is transformed into the problem of coefficient optimization, which greatly simplifies the process of constructing the multi-kernel function, and the sliding time window optimized multi-kernel LS-SVM packet dropout online compensation can ensure high compensation accuracy. The adverse effect of packet dropout on the control system is greatly reduced.

2. The neural network PID approach law sliding-mode control can not only guarantee the fast response speed, but also reduce the chattering amplitude respectively compared with the other approach law sliding-mode control, which shows that it has made great improvement in reducing the chattering, and both the response speed and the chattering suppression are taken into consideration.

**Table 4. Control chattering results of fuel supply response.**

| approach law | Response time(s) | Steady-state error(%) |
|---|---|---|
| Fixed parameter PID | 5.2 | 7.78 |
| Exponential | 11.3 | 8.96 |
| Global | 14.8 | 1.43 |
| Segment | 4.8 | 5.86 |
| Fuzzy exponential | 5.1 | 2.18 |
| RBF-PID | 3.9 | 0.53 |

3. For the linear model with small deviation, the neural network PID sliding-mode control based on the sliding time window multi-kernel LS-SVM online compensation can better realize the tracking control of the multi-packet transmission aeroengine network control system with time-delay and packet dropout, and has certain robustness to the value of the packet dropout rate. For the nonlinear model, further verification is needed.

Because the small deviation state space model of aeroengine nominal point is used in this paper, the aeroengine control changing in the whole envelope range needs to be further studied; in addition, for the network control system, there are many assumptions of ideal state, thus, the control effect of the packet timing disorder, network scheduling algorithm, etc. shall be considered as well in the next research.

## Supporting information

**S1 Data.**
(ZIP)

## Acknowledgments

The author would like to thank the editor, associate editor, and anonymous reviewers for their constructive comments.

## Author Contributions

**Conceptualization:** Li Guangfu.

**Data curation:** Li Guangfu.

**Formal analysis:** Li Guangfu.

**Funding acquisition:** Li Guangfu, Wang Xu.

**Investigation:** Li Guangfu.

**Methodology:** Li Guangfu.

**Project administration:** Li Guangfu.

**Resources:** Li Guangfu, Wang Xu.

**Software:** Li Guangfu.

**Supervision:** Li Guangfu, Wang Xu, Ren Jia.

**Validation:** Li Guangfu, Ren Jia.

**Visualization:** Li Guangfu, Ren Jia.

**Writing – original draft:** Li Guangfu.

**Writing – review & editing:** Li Guangfu, Wang Xu, Ren Jia.

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
