## [Decision Letter · Decision Letter 0]

31 Mar 2020

PONE-D-20-03888

Multiple-packet Transmission Aero-engine DCS Neural Network Sliding Mode Control Based on Multi-kernel LS-SVM Packet Dropout Online Compensation

PLOS ONE

Dear Mr Guangfu,

Thank you for submitting your manuscript to PLOS ONE. After careful consideration, we feel that it has merit but does not fully meet PLOS ONE’s publication criteria as it currently stands. Therefore, we invite you to submit a revised version of the manuscript that addresses the points raised during the review process.

We would appreciate receiving your revised manuscript by May 15 2020 11:59PM. To enhance the reproducibility of your results, we recommend that if applicable you deposit your laboratory protocols in protocols.io, where a protocol can be assigned its own identifier (DOI) such that it can be cited independently in the future. For instructions see: http://journals.plos.org/plosone/s/submission-guidelines#loc-laboratory-protocols

We look forward to receiving your revised manuscript.

Kind regards,

Yanzheng Zhu

Academic Editor

PLOS ONE

Journal Requirements:

2. Please amend either the abstract on the online submission form (via Edit Submission) or the abstract in the manuscript so that they are identical.

Additional Editor Comments (if provided):

Based on the two reviewers' comments, the authors should be revised the paper very carefully, and respond the concerns one by one. Besides，the writting is very confused and lack of careful.

Reviewers' comments:

Reviewer's Responses to Questions

**Comments to the Author**

1. Is the manuscript technically sound, and do the data support the conclusions?

Reviewer #1: Yes

Reviewer #2: Partly

2. Has the statistical analysis been performed appropriately and rigorously? 

Reviewer #1: Yes

Reviewer #2: Yes

3. Have the authors made all data underlying the findings in their manuscript fully available?

Reviewer #1: Yes

Reviewer #2: Yes

4. Is the manuscript presented in an intelligible fashion and written in standard English?

Reviewer #1: Yes

Reviewer #2: No

5. Review Comments to the Author

Reviewer #1: In this paper, a sliding mode control method is proposed for the Aero-engine DCS with induced delay and random packet dropout. The multi-kernel function is used to design the online predictive compensation, then the neural network PID approach with sliding-mode control is proposed to reduce the chattering, which is robustness to the value of the packet dropout rate. In general, the presentation of the paper seems acceptable, and the equations, diagrams, and tables seem appropriate and clear. However, there are some issues for the authors to consider to improve the overall quality of the manuscript.

1) For the Aero-engine DCS system, what are the main advantages of the proposed method over the existing control methods?

2) In (19), how to determine the proportionality coefficient, integral coefficient, differential coefficient in the sliding mode control, since the choosing of these constants are important to the control performance?

3) What is the difference between the LS-SVM and traditional SVM methods? Why you choose LS-SVM in your design?

4) What are the complexity issues in your controller design?

5) The reference list has been relatively comprehensive in terms of the context of this paper. However, the following literature about the NCS design might be of relevance to some extent, such as HMM-based H-infinity filtering for discrete-time Markov jump LPV systems over unreliable communication channels; Adaptive fuzzy sliding mode control for network-based nonlinear systems with actuator failures; Observer-based control for cyber-physical systems with periodic dos attacks via a cyclic switching strategy.

6) Based on the topic addressed in this paper, the authors are suggested to propose some relevant topics for future work.

Reviewer #2: The reviewer got confused after reading the article. In Conclusion (3), the authors seemed to use the linear model to complete the analysis while so-called strong nonlinear characteristics were all across the paper. Plus, many details in the paper have shown the unprofessionalism. Just to mention a few:

1. Just a half sentence in the Abstract field sent to the reviewer, shown as “In the view of the nonlinear characteristics of the”

2. On page 22, “as the red curve in Figure 5”. Double check if it is Figure 5 or not.

3. Misuse of “Figure X, Fig X”.

4. On Page 13, the labels of the three functions were overlapped by the text.

5. Double check the Y-axis of all the figures

6. PLOS authors have the option to publish the peer review history of their article (what does this mean?). If published, this will include your full peer review and any attached files.

Reviewer #1: No

Reviewer #2: No

---

## [Author Response · Author response to Decision Letter 0]

1 May 2020

Response to Reviewers

Dear Editor:

On behalf of my co-authors, we thank you very much for giving us an opportunity to revise our manuscript, we appreciate editor and reviewers very much for their positive and constructive comments and suggestions on our manuscript entitled “Multiple-packet Transmission Aero-engine DCS Neural Network Sliding Mode Control Based on Multi-kernel LS-SVM Packet Dropout Online Compensation”. (ID: PONE-D-20-03888).Those comments are all valuable and very helpful for revising and improving our paper, as well as the important guiding significance to our researches. We have studied comments carefully and have made correction which we hope meet with approval. Revised portion are marked in red in the paper. The main corrections in the paper and the responds to the reviewer’s comments are as flowing:

Responds to the reviewer’s comments:

Reviewer #1:

1. For the Aero-engine DCS system, what are the main advantages of the proposed method over the existing control methods?

Response: For the aeroengine DCS system, the main advantages of the neural network PID approach law sliding mode controller are as follows:

1. The traditional sliding mode control does not consider the delay in the distributed control system, the impact of packet loss on the control effect, but also ignores the existence of multi packet transmission. The method proposed in this paper takes into account a variety of situations, effectively offsets the impact of delay packet loss and multi packet transmission, and achieves effective control.

2. For the packet loss compensation method, the compensation method proposed in this paper effectively realizes the prediction compensation of multi-core LS-SVM, and uses chaos adaptive artificial fish swarm optimization algorithm to transform the construction of multi-core functions into optimization problems, and realizes the optimal compensation of multi-core LS-SVM.

3. Compared with other types of approach law sliding mode control, neural network PID approach law sliding mode control can adjust the three parameters of PID adaptively, ensure the convergence speed more effectively, and suppress chattering better, which has better effect on aeroengine control.

2. In (19), how to determine the proportionality coefficient, integral coefficient, differential coefficient in the sliding mode control, since the choosing of these constants are important to the control performance?

Response: The initial parameters value of the sliding-mode controller is set as proportional coefficient , integral coefficient , differential coefficient.The input of the neural network is the sliding mode switching function and its variation. The two inputs can reflect the state of the sliding mode surface and the future movement trend. The output is the three parameters of the PID approach law. When the input is close to the sliding mode surface and the change of the sliding mode switching function is large, the proportion coefficient should be reduced, the integral coefficient increased, the chattering amplitude value reduced and the stability quickly achieved When the sliding surface is far away from the sliding surface, the proportion coefficient should be increased, and the integral coefficient and differential coefficient should be reduced to achieve the effect of fast tightening the sliding surface. When the sliding surface is reached, the integral coefficient and buffeting frequency should be reduced. Radial basis function neural network belongs to multilayer feedforward neural network, which has strong nonlinear mapping ability. Thus, it effectively realizes the nonlinear mapping of switching function and its variation to three parameters of PID reaching law.

3.What is the difference between the LS-SVM and traditional SVM methods? Why you choose LS-SVM in your design?

Response: Difference between LS-SVM and traditional SVM:

(1)LS-SVM uses equality constraint, while traditional SVM is inequality constraint;

(2)LS-SVM uses equality constraints on each sample point, so it does not impose any constraints on the relaxation vector, which is also an important reason for LSSVM to lose sparsity;

(3)LS-SVM simplifies the problem further by solving the equality constraint and the least square problem.

In this paper, although it is a non-linear problem, it can still be solved by the mode of linear equation, while LS-SVM is faster and easier to meet the solution conditions when dealing with linear equation, so LS-SVM is used.

4.What are the complexity issues in your controller design?

Response: The complexity problem mainly includes the uncertainty and randomness of packet loss in the process of multi packet transmission, as well as the nonlinearity of control object. In addition, for the sliding mode control, how to design the sliding mode approach law with fast control speed and small buffeting amplitude frequency is also a complex factor in the controller design.

5.The reference list has been relatively comprehensive in terms of the context of this paper. However, the following literature about the NCS design might be of relevance to some extent, such as HMM-based H-infinity filtering for discrete-time Markov jump LPV systems over unreliable communication channels; Adaptive fuzzy sliding mode control for network-based nonlinear systems with actuator failures; Observer-based control for cyber-physical systems with periodic dos attacks via a cyclic switching strategy.

Response:Thank you for your valuable suggestion. These articles have been added to the references.

6.Based on the topic addressed in this paper, the authors are suggested to propose some relevant topics for future work.

Response: Because the model used in this paper is the small deviation state space model of aeroengine nominal point, the aeroengine control which changes in the whole envelope range needs to be further studied; in addition, for the network control system, there are many assumptions of ideal state, the next research also needs to consider the control effect of the packet timing disorder, network scheduling algorithm, etc Influence.

The relevant outlook has been supplemented in the conclusion.

Reviewer #2:

1. Just a half sentence in the Abstract field sent to the reviewer, shown as “In the view of the nonlinear characteristics of the”

Response: Thank you for your valuable comments. The working environment of the engine is complex, the working condition is bad, there are inevitably uncertain factors such as parameter perturbation and external interference, so the model presents strong nonlinear characteristics. Aiming at the nonlinear model and the current aeroengine controller design, the commonly used method is to divide the flight envelope into several sub regions that meet certain performance indexes. In each region, a representative nominal work point is selected, and a small deviation state space model of the point is established. The controller of the current envelope region is designed for the model. Therefore, the design of the controller is studied by using the linear small deviation state space model. The relevant description has been supplemented in the article..

2. On page 22, “as the red curve in Figure 5”. Double check if it is Figure 5 or not.

Response: I am sorry for my carelessness. We have changed it.

3. Misuse of “Figure X, Fig X”.

Response: I am sorry for my carelessness. We have changed it to “Fig X”.

4.On Page 13, the labels of the three functions were overlapped by the text.

Response: I am sorry for my carelessness. We have changed labels.

5.Double check the Y-axis of all the figures.

Response: I am sorry for my carelessness. We have checked the Y-axis of all the figures and have changed the description of Y-axis, such as units and data representation.

 In addition, we have also made a detailed revision of the grammar and words used in the language expression of the article.We tried our best to improve the manuscript and made some changes in the manuscript. These changes will not influence the content and framework of the paper. And here we did not list the changes but marked in red in revised paper.

We appreciate for Editors/Reviewers’ warm work earnestly, and hope that the correction will meet with approval.

Once again, thank you very much for your comments and suggestions.

Looking forward to hearing from you.

Thank you and best regards.

Yours sincerely,

Guangfu Li

---

## [Decision Letter · Decision Letter 1]

26 May 2020

Multi-packet Transmission Aero-engine DCS Neural Network Sliding Mode Control Based on Multi-Kernel LS-SVM Packet Dropout  Online Compensation

PONE-D-20-03888R1

Dear Dr. Guangfu,

We are pleased to inform you that your manuscript has been judged scientifically suitable for publication and will be formally accepted for publication once it complies with all outstanding technical requirements.

With kind regards,

Yanzheng Zhu

Academic Editor

PLOS ONE

Additional Editor Comments (optional):

Based on the reviewers' suggestions, the paper currently can be accepted for publication now. Please polish the whole paper again before submitting the final version.

Reviewers' comments:

Reviewer's Responses to Questions

**Comments to the Author**

1. If the authors have adequately addressed your comments raised in a previous round of review and you feel that this manuscript is now acceptable for publication, you may indicate that here to bypass the “Comments to the Author” section, enter your conflict of interest statement in the “Confidential to Editor” section, and submit your "Accept" recommendation.

Reviewer #1: (No Response)

Reviewer #2: All comments have been addressed

2. Is the manuscript technically sound, and do the data support the conclusions?

Reviewer #1: Yes

Reviewer #2: Yes

3. Has the statistical analysis been performed appropriately and rigorously? 

Reviewer #1: Yes

Reviewer #2: Yes

4. Have the authors made all data underlying the findings in their manuscript fully available?

Reviewer #1: Yes

Reviewer #2: Yes

5. Is the manuscript presented in an intelligible fashion and written in standard English?

Reviewer #1: Yes

Reviewer #2: Yes

6. Review Comments to the Author

Reviewer #1: (No Response)

Reviewer #2: The reviewer is happy with the revised version provieded by the authors. The paper is ready to be accepted.

7. PLOS authors have the option to publish the peer review history of their article (what does this mean?). If published, this will include your full peer review and any attached files.

Reviewer #1: No

Reviewer #2: No

---

## [Editor Report · Acceptance letter]

5 Jun 2020

PONE-D-20-03888R1 

Multi-packet Transmission Aero-engine DCS Neural Network Sliding Mode Control Based on Multi-Kernel LS-SVM Packet Dropout  Online Compensation 

Dear Dr. Guangfu:

I'm pleased to inform you that your manuscript has been deemed suitable for publication in PLOS ONE. Congratulations! Your manuscript is now with our production department. 

Kind regards, 

on behalf of

Dr. Yanzheng Zhu 

Academic Editor

PLOS ONE